# Learning Graph Models for Retrosynthesis Prediction

**Vignesh Ram Somnath**[1]       **Charlotte Bunne**[1]       **Connor W. Coley**[2]

**Andreas Krause**[1]      **Regina Barzilay**[3]

[1]Department of Computer Science, ETH
[2]Department of Chemical Engineering, MIT
[3]Computer Science and Artificial Intelligence Lab, MIT
[1]{vsomnath, bunnec, krausea}@ethz.ch, [2]ccoley@mit.edu, [3]regina@csail.mit.edu

## Abstract

Retrosynthesis prediction is a fundamental problem in organic synthesis, where the task is to identify precursor molecules that can be used to synthesize a target molecule. A key consideration in building neural models for this task is aligning model design with strategies adopted by chemists. Building on this viewpoint, this paper introduces a graph-based approach that capitalizes on the idea that the graph topology of precursor molecules is largely unaltered during a chemical reaction. The model first predicts the set of graph edits transforming the target into incomplete molecules called *synthons*. Next, the model learns to expand synthons into complete molecules by attaching relevant *leaving groups*. This decomposition simplifies the architecture, making its predictions more interpretable, and also amenable to manual correction. Our model achieves a top-1 accuracy of 53.7%, outperforming previous template-free and semi-template-based methods.

## 1   Introduction

Retrosynthesis prediction, first formalized by E. J. Corey [Corey, 1991] is a fundamental problem in organic synthesis that attempts to identify a series of chemical transformations for synthesizing a target molecule. In the single-step formulation, the task is to identify a set of reactant molecules given a target. Beyond simple reactions, many practical tasks involving complex organic molecules are difficult even for expert chemists. As a result, substantial experimental exploration is needed to cover for deficiencies of analytical approaches. This has motivated interest in computer-assisted retrosynthesis [Corey and Wipke, 1969], with a recent surge in machine learning methods [Chen et al., 2019, Coley et al., 2017b, Dai et al., 2019, Zheng et al., 2019, Genheden et al., 2020].

Computationally, the main challenge is how to explore the combinatorial space of reactions that can yield the target molecule. Largely, previous methods for retrosynthesis prediction can be divided into template-based [Coley et al., 2017b, Dai et al., 2019, Segler and Waller, 2017] and template-free [Chen et al., 2019, Zheng et al., 2019] approaches. Template-based methods match a target molecule against a large set of templates, which are molecular subgraph patterns that highlight changes during a chemical reaction. Despite their interpretability, these methods fail to generalize to new reactions. Template-free methods bypass templates by learning a direct mapping from the SMILES [Weininger, 1988] representations of the product to reactants. Despite their greater generalization potential, these methods generate reactant SMILES character by character, increasing generation complexity.

**a Edit Prediction**

**b Synthon Completion**

Figure 1: **Overview of Our Approach**. **a**. **Edit Prediction**. We train a model to learn a distribution over possible graph edits. In this case, the correct edit corresponds to breaking the bond marked in red. Applying this edit produces two synthons. **b**. **Synthon Completion**. Another model is trained to pick candidate leaving groups (blue) for each synthon from a discrete vocabulary, which are then attached to produce the final reactants.

Another important consideration in building retrosynthesis models is aligning model design with strategies adopted by expert chemists. These strategies are influenced by fundamental properties of chemical reactions, independent of complexity level: (i.) the product atoms are always a subset of the reactant atoms[1], and (ii.) the molecular graph topology is largely unaltered from products to reactants. For example, in the standard retrosynthesis dataset, only 6.3% of the atoms in the product undergo any change in connectivity.

This consideration has received more attention in recent semi-template-based methods [Shi et al., 2020, Yan et al., 2020], that generate reactants from a product in two stages: (i.) first identify intermediate molecules called synthons, (ii.) and then complete synthons into reactants by sequential generation of atoms or SMILES characters.. Our model GRAPHRETRO also uses a similar workflow. However, we avoid sequential generation for completing synthons by instead selecting subgraphs called *leaving groups* from a precomputed vocabulary. This vocabulary is constructed during preprocessing by extracting subgraphs that differ between a synthon and the corresponding reactant. The vocabulary has a small size (170 for USPTO-50k) indicating remarkable redundancy, while covering 99.7% of the test set. Operating at the level of these subgraphs greatly reduces the complexity of reactant generation, with improved empirical performance. This formulation also simplifies our architecture, and makes our predictions more transparent, interpretable and amenable to manual correction.

The benchmark dataset for evaluating retrosynthesis models is USPTO-50k [Schneider et al., 2016], which consists of 50000 reactions across 10 reaction classes. The dataset contains an unexpected shortcut towards predicting the edit, in that the product atom with atom-mapping 1 is part of the edit in 75% of the cases, allowing predictions that depend on the position of the atom to overestimate performance. We canonicalize the product SMILES and remap the existing dataset, thereby removing the shortcut. On this remapped dataset, GRAPHRETRO achieves a top-1 accuracy of 53.7% when the reaction class is not known, outperforming both template-free and semi-template-based methods.

## 2 Related Work

**Retrosynthesis Prediction** Existing machine learning methods for retrosynthesis prediction can be divided into template-based, template-free and recent semi-template-based approaches.

*Template-Based*: Templates are either hand-crafted by experts [Hartenfeller et al., 2011, Szymkuć et al., 2016], or extracted algorithmically from large databases Coley et al. [2017a], Law et al. [2009]. Exhaustively applying large template sets is expensive due to the involved subgraph

---

[1]ignoring impurities

matching procedure. Template-based methods therefore utilize different ways of prioritizing templates, by either learning a conditional distribution over the template set [Segler and Waller, 2017], ranking templates based on molecular similarities to precedent reactions [Coley et al., 2017b] or directly modelling the joint distribution of templates and reactants using logic variables [Dai et al., 2019]. Despite their interpretability, these methods fail to generalize outside their rule set.

*Template-Free*: Template-free methods [Liu et al., 2017, Zheng et al., 2019, Chen et al., 2019] learn a direct transformation from products to reactants using architectures from neural machine translation and a string based representation of molecules called SMILES [Weininger, 1988]. Linearizing molecules as strings does not utilize the inherently rich chemical structure. In addition, the reactant SMILES are generated from scratch, character by character. Attempts have been made to improve validity by adding a syntax corrector [Zheng et al., 2019] and a mixture model to improve diversity of suggestions [Chen et al., 2019], but the performance remains worse than [Dai et al., 2019] on the standard retrosynthesis dataset. Sun et al. [2021] formulate retrosynthesis using energy-based models, with additional parameterizations and loss terms to enforce the duality between forward (reaction prediction) and backward (retrosynthesis) prediction.

*Semi-Template-Based*: Our work is closely related to recently proposed semi-template-based methods [Shi et al., 2020, Yan et al., 2020], which first identify synthons and then expand synthons into reactants through sequential generation using either a graph generative model [Shi et al., 2020] or a Transformer [Yan et al., 2020]. To reduce the complexity of reactant generation, we instead complete synthons using subgraphs called *leaving groups* selected from a precomputed vocabulary. This allows us to view synthon completion as a classification problem instead of a generative one. We also utilize the dependency graph between possible edits, and update edit predictions using a *message passing network* (MPN) [Gilmer et al., 2017] on this graph. Both innovations together yield a 4.8% and 3.3% performance improvement respectively over previous semi-template-based methods.

**Reaction Center Identification**    The reaction center covers a small number of participating atoms involved in the reaction. Our work is also related to models that predict reaction outcomes by learning to rank atom pairs based on their likelihood to be in the reaction center [Coley et al., 2019, Jin et al., 2017]. The task of identifying the reaction center is related to the step of deriving the synthons in our formulation. Our work departs from [Coley et al., 2019, Jin et al., 2017] as we utilize the property that new bond formations occur rarely (~0.1%) from products to synthons, allowing us to predict a score only for existing bonds and atoms and reduce prediction complexity from $O(N^2)$ to $O(N)$. We also utilize the dependency graph between possible edits, and update edit predictions using a MPN on this graph.

**Utilizing Substructures**    Substructures have been utilized in various tasks from sentence generation by fusing phrases to molecule generation and optimization [Jin et al., 2018, 2020]. Our work is closely related to [Jin et al., 2020] which uses precomputed substructures as building blocks for property-conditioned molecule generation. However, instead of precomputing, synthons —analogous building blocks for reactants— are indirectly learnt during training.

## 3    Model Design

Our approach leverages the property that graph topology is largely unaltered from products to reactants. To achieve this, we first derive suitable building blocks from the product called *synthons*, and then complete them into valid reactants by adding specific functionalities called *leaving groups*. These derivations, called *edits*, are characterized by modifications to bonds or hydrogen counts on atoms. We first train a neural network to predict a score for possible edits (Section 3.1). The edit with the highest score is then applied to the product to obtain synthons. Since the number of unique leaving groups are small, we model leaving group selection as a classification problem over a precomputed vocabulary (Section 3.2). To produce candidate reactants, we attach the predicted leaving group to the corresponding synthon through chemically constrained rules. The overall process is outlined in Figure 1. Before describing the two modules, we introduce relevant preliminaries that set the background for the remainder of the paper.

**Retrosynthesis Prediction**   A retrosynthesis pair $R$ is described by a pair of molecular graphs $(\mathcal{G}_p, \mathcal{G}_r)$, where $\mathcal{G}_p$ are the products and $\mathcal{G}_r$ the reactants. A molecular graph is described as $\mathcal{G} = (\mathcal{V}, \mathcal{E})$ with atoms $\mathcal{V}$ as nodes and bonds $\mathcal{E}$ as edges. Prior work has focused on the single product case, while reactants can have multiple connected components, i.e. $\mathcal{G}_r = \{\mathcal{G}_{r_c}\}_{c=1}^C$. Retrosynthesis pairs are *atom-mapped* so that each product atom has a unique corresponding reactant atom. The retrosynthesis task then, is to infer $\{\mathcal{G}_{r_c}\}_{c=1}^C$ given $\mathcal{G}_p$.

**Edits**   Edits consist of (i.) atom pairs $\{(a_i, a_j)\}$ where the bond type changes from products to reactants, and (ii.) atoms $\{a_i\}$ where the number of hydrogens attached to the atom change from products to reactants. We denote the set of edits by $E$. Since retrosynthesis pairs in the training set are atom-mapped, edits can be automatically identified by comparing the atoms and atom pairs in the product to their corresponding reactant counterparts.

**Synthons and Leaving Groups**   Applying edits $E$ to the product $\mathcal{G}_p$ results in incomplete molecules called *synthons*. Synthons are analogous to rationales or building blocks, which are expanded into valid reactants by adding specific functionalities called *leaving groups* that are responsible for its reactivity. We denote synthons by $\mathcal{G}_s$ and leaving groups by $\mathcal{G}_l$. We further assume that synthons and leaving groups have the same number of connected components as the reactants, i.e $\mathcal{G}_s = \{\mathcal{G}_{s_c}\}_{c=1}^C$ and $\mathcal{G}_l = \{\mathcal{G}_{l_c}\}_{c=1}^C$. This assumption holds for 99.97% reactions in the training set.

Formally, our model generates reactants by first predicting the set of edits $E$ that transform $\mathcal{G}_p$ into $\mathcal{G}_s$, followed by predicting a leaving group $\mathcal{G}_{l_c}$ to attach to each synthon $\mathcal{G}_{s_c}$. The model is defined as

$$P(\mathcal{G}_r|\mathcal{G}_p) = \sum_{E, \mathcal{G}_l} P(E|\mathcal{G}_p)P(\mathcal{G}_l|\mathcal{G}_p, \mathcal{G}_s), \tag{1}$$

where $\mathcal{G}_s, \mathcal{G}_r$ are deterministic given $E, \mathcal{G}_l$, and $\mathcal{G}_p$.

## 3.1   Edit Prediction

For a given retrosynthesis pair $R = (\mathcal{G}_p, \mathcal{G}_r)$, we predict an edit score only for existing bonds and atoms, instead of every atom pair as in [Coley et al., 2019, Jin et al., 2017]. This choice is motivated by the low frequency (~0.1%) of new bond formations in the training set examples. Coupled with the sparsity of molecular graphs, this reduces the prediction complexity from $O(N^2)$ to $O(N)$ for a product with $N$ atoms. Our edit prediction model has variants tailored to single and multiple edit prediction. Since 95% of the training set consists of single edit examples, the remainder of this section describes the setup for single edit prediction. A detailed description of our multiple edit prediction model can be found in Appendix **??**.

Each bond $(u, v)$ in $\mathcal{G}_p$ is associated with a label $y_{uvk} \in \{0, 1\}$ indicating whether its bond type $k$ has changed from the products to reactants. Each atom $u$ is associated with a label $y_u \in \{0, 1\}$ indicating a change in hydrogen count. We predict edit scores using representations that are learnt using a graph encoder.

**Graph Encoder**   To obtain atom representations, we use a variant of the *message passing network* (MPN) described in [Gilmer et al., 2017]. Each atom $u$ has a feature vector $\mathbf{x}_u$ indicating its atom type, degree and other properties. Each bond $(u, v)$ has a feature vector $\mathbf{x}_{uv}$ indicating its aromaticity, bond type and ring membership. For simplicity, we denote the encoding process by $\mathrm{MPN}(\cdot)$ and describe architectural details in Appendix **??**. The MPN computes atom representations $\{\mathbf{c}_u | u \in \mathcal{G}\}$ via

$$\{\mathbf{c}_u\} = \mathrm{MPN}(\mathcal{G}, \{\mathbf{x}_u\}, \{\mathbf{x}_{uv}\}_{v \in \mathcal{N}(u)}), \tag{2}$$

where $\mathcal{N}(u)$ denotes the neighbors of atom $u$. The graph representation $\mathbf{c}_\mathcal{G}$ is an aggregation of atom representations, i.e. $\mathbf{c}_\mathcal{G} = \sum_{u \in \mathcal{V}} \mathbf{c}_u$. When $\mathcal{G}$ has connected components $\{\mathcal{G}_i\}$, we get a set of graph representations $\{\mathbf{c}_{\mathcal{G}_i}\}$. For a bond $(u, v)$, we define its representation $\mathbf{c}_{uv} = (\mathrm{ABS}(\mathbf{c}_u, \mathbf{c}_v) || \mathbf{c}_u + \mathbf{c}_v)$, where ABS denotes absolute difference and $||$ refers to concatenation. This ensures our representations are permutation invariant. These representations are then used to predict atom and bond edit scores using corresponding neural networks,

$$s_u = \mathbf{u_a}^T \tau(\mathbf{W_a c}_u + b) \tag{3}$$

$$s_{uvk} = \mathbf{u_k}^T \tau(\mathbf{W_k c}_{uv} + b_k), \tag{4}$$

where $\tau(\cdot)$ is the ReLU activation function.

**Updating Bond Edit Scores**  Unlike a typical classification problem where the labels are independent, edits can have possible dependencies between each other. For example, bonds part of a stable system such as an aromatic ring have a greater tendency to remain unchanged (label 0). We attempt to leverage such dependencies to update initial edit scores. To this end, we build a graph with bonds $(u, v)$ as nodes, and introduce an edge between bonds sharing an atom. We use another $\text{MPN}(\cdot)$ on this graph to learn aggregated neighborhood messages $\mathbf{m}_{uv}$, and update the edit scores $s_{uvk}$ in a manner similar to how LSTMs update representations,

$$f_{uvk} = \sigma(\mathbf{W_{kx}^f x}_{uv} + \mathbf{W_{km}^f m}_{uv}) \tag{5}$$

$$i_{uvk} = \sigma(\mathbf{W_{kx}^i x}_{uv} + \mathbf{W_{km}^i m}_{uv}) \tag{6}$$

$$\tilde{m}_{uvk} = \mathbf{u_m}\tau(\mathbf{W_{kx}^m x}_{uv} + \mathbf{W_{km}^m m}_{uv}) \tag{7}$$

$$\tilde{s}_{uvk} = f_{uvk} \cdot s_{uvk} + i_{uvk} \cdot \tilde{m}_{uvk}. \tag{8}$$

**Training**  We train by minimizing the cross-entropy loss over possible bond and atom edits

$$\mathcal{L}_e = -\sum_{(\mathcal{G}_p, E)} \left( \sum_{((u,v),k) \in E} y_{uvk} \log(\tilde{s}_{uvk}) + \sum_{u \in E} y_u \log(s_u) \right). \tag{9}$$

The cross-entropy loss enforces the model to learn a distribution over possible edits instead of reasoning about each edit independently, as with the binary cross entropy loss used in [Jin et al., 2017, Coley et al., 2019].

## 3.2  Synthon Completion

Synthons are completed into valid reactants by adding specific functionalities called *leaving groups*. This involves two complementary tasks: (i.) selecting the appropriate leaving group, and (ii.) attaching the leaving group to the synthon. As ground truth leaving groups are not directly provided, we extract the leaving groups and construct a vocabulary $\mathcal{X}$ of unique leaving groups during preprocessing.

The vocabulary has a limited size ($|\mathcal{X}| = 170$ for a standard dataset with $50,000$ examples, and $72000$ synthons) indicating the redundancy of leaving groups used in accomplishing retrosynthetic transformations. This redundancy also allows us to formulate leaving group selection as a classification problem over $\mathcal{X}$, while retaining the ability to generate diverse reactants using different combinations of leaving groups.

**Vocabulary Construction**  Before constructing the vocabulary, we align connected components of synthon and reactant graphs by comparing atom mapping overlaps. Using aligned pairs $\mathcal{G}_{s_c} = (\mathcal{V}_{s_c}, \mathcal{E}_{s_c})$ and $\mathcal{G}_{r_c} = (\mathcal{V}_{r_c}, \mathcal{E}_{r_c})$ as input, the leaving group vocabulary $\mathcal{X}$ is constructed by extracting subgraphs $\mathcal{G}_{l_c} = (\mathcal{V}_{l_c}, \mathcal{E}_{l_c})$ such that $\mathcal{V}_{l_c} = \mathcal{V}_{r_c} \setminus \mathcal{V}_{s_c}$. Atoms $\{a_i\}$ in the leaving groups that attach to synthons are marked with a special symbol. We also add three tokens to $\mathcal{X}$ namely START, which indicates the start of synthon completion, END, which indicates that there is no leaving group to add and PAD, which is used to handle variable numbers of synthon components in a minibatch.

**Leaving Group Selection**  For synthon component $c \leq C$, where $C$ is the number of connected components in the synthon graph, we use three inputs for leaving group selection – the product representation $\mathbf{c}_{\mathcal{G}_p}$, the synthon component representation $\mathbf{c}_{\mathcal{G}_{s_c}}$, and the leaving group representation for the previous synthon component, $\mathbf{e}_{l_c-1}$. The product and synthon representations are learnt using the $\text{MPN}(\cdot)$. For each $x_i \in \mathcal{X}$, representations can be learnt by either training independent embedding vectors (*ind*) or by treating each $x_i$ as a subgraph and using the $\text{MPN}(\cdot)$ (*shared*). In the *shared* setting, we use the same $\text{MPN}(\cdot)$ as the product and synthons.

The leaving group probabilities are then computed by combining $\mathbf{c}_{\mathcal{G}_p}$, $\mathbf{c}_{\mathcal{G}_{s_c}}$ and $\mathbf{e}_{l_{c-1}}$ via a single layer neural network and $\mathrm{softmax}$ function

$$\hat{q}_{l_c} = \mathrm{softmax}\left(\mathbf{U}\tau\left(\mathbf{W_1}\mathbf{c}_{\mathcal{G}_p} + \mathbf{W_2}\mathbf{c}_{\mathcal{G}_{s_c}} + \mathbf{W_3}\mathbf{e}_{l_{(c-1)}}\right)\right), \tag{10}$$

where $\hat{q}_{l_c}$ is distribution learnt over $\mathcal{X}$. Using the representation of the previous leaving group $\mathbf{e}_{l_{c-1}}$ allows the model to understand *combinations* of leaving groups that generate the desired product from the reactants. We also include the product representation $\mathbf{c}_{\mathcal{G}_p}$ as the synthon graphs are derived from the product graph.

**Training**   For step $c$, given the one hot encoding of the true leaving group $q_{l_c}$, we minimize the cross-entropy loss

$$\mathcal{L}_s = \sum_{c=1}^{C} \mathcal{L}(\hat{q}_{l_c}, q_{l_c}). \tag{11}$$

Training utilizes teacher-forcing [Williams and Zipser, 1989] so that the model makes predictions given correct histories. During inference, at every step, we use the representation of leaving group from the previous step with the highest predicted probability.

**Leaving Group Attachment**   Attaching leaving groups to synthons is a deterministic process and not learnt during training. The task involves identification of the type of bonds to add between attaching atoms in the leaving group (marked during vocabulary construction), and the atom(s) participating in the edit. These bonds can be inferred by applying the *valency* constraint, which determines the maximum number of neighbors for each atom. The attachment process does not modify any stereochemistry. Given synthons and leaving groups, the attachment process has a 100% accuracy. The detailed procedure is described in Appendix **??**.

### 3.3   Inference

Inference is performed using beam search with a log-likelihood scoring function. For a beam width $n$, we select $n$ edits with highest scores and apply them to the product to obtain $n$ synthons, where each synthon can consist of multiple connected components. The synthons form the nodes for beam search. Each node maintains a cumulative score by aggregating the log-likelihoods of the edit and predicted leaving groups. Leaving group inference starts with a connected component for each synthon, and selects $n$ leaving groups with highest log-likelihoods. From the $n^2$ possibilities, we select $n$ nodes with the highest cumulative scores. This process is repeated until all nodes have a leaving group predicted for each synthon component.

## 4   Evaluation

Evaluating retrosynthesis models is challenging as multiple sets of reactants can be generated from the same product. To deal with this, previous works [Coley et al., 2017b, Dai et al., 2019] evaluate the ability of the model to recover retrosynthetic strategies recorded in the dataset.

**Data**   We use the benchmark dataset USPTO-50k [Schneider et al., 2016] for all our experiments. The dataset contains $50,000$ atom-mapped reactions across 10 reaction classes. We use the same dataset version and splits as provided by [Dai et al., 2019]. The USPTO-50k dataset contains a shortcut in that the product atom with atom-mapping 1 is part of the edit in ~75% of the cases. If the product SMILES is not canonicalized, predictions utilizing operations that depend on the position of the atom or bond will be able to use the shortcut, and overestimate performance. We canonicalize the product SMILES, and reassign atom-mappings to the reactant atoms based on the canonical ordering, which removes the shortcut. Details on the remapping procedure can be found in Appendix **??**.

**Evaluation**   We use the top-$n$ accuracy ($n = 1, 3, 5, 10$) as our evaluation metric, defined as the fraction of examples where the recorded reactants are suggested by the model with rank $\leq n$. Following prior work [Coley et al., 2017b, Zheng et al., 2019, Dai et al., 2019], we compute the accuracy

Table 1: **Top-$n$ exact match accuracy**. Best values within each section are highlighted in bold.

| Model | $n =$ | Top-$n$ Accuracy (%) | | | | | | | |
|---|---|---|---|---|---|---|---|---|---|
| | | Reaction class known | | | | Reaction class unknown | | | |
| | | 1 | 3 | 5 | 10 | 1 | 3 | 5 | 10 |
| **Template-Based** | | | | | | | | | |
| RETROSIM [Coley et al., 2017b] | | 52.9 | 73.8 | 81.2 | 88.1 | 37.3 | 54.7 | 63.3 | 74.1 |
| NEURALSYM [Segler and Waller, 2017] | | 55.3 | 76.0 | 81.4 | 85.1 | 44.4 | 65.3 | 72.4 | 78.9 |
| GLN [Dai et al., 2019] | | 64.2 | 79.1 | 85.2 | 90.0 | 52.5 | 69.0 | 75.6 | 83.7 |
| DUALTB [Sun et al., 2021] | | **67.7** | **84.8** | **88.9** | **92.0** | **55.2** | **74.6** | **80.5** | **86.9** |
| **Template-Free** | | | | | | | | | |
| SCROP [Zheng et al., 2019] | | 59.0 | 74.8 | 78.1 | 81.1 | 43.7 | 60.0 | 65.2 | 68.7 |
| LV-TRANSFORMER [Chen et al., 2019] | | - | - | - | - | 40.5 | 65.1 | 72.8 | 79.4 |
| DUALTF [Sun et al., 2021] | | **65.7** | **81.9** | **84.7** | **85.9** | **53.6** | **70.7** | **74.6** | **77.0** |
| **Semi-Template-Based** | | | | | | | | | |
| G2GS [Shi et al., 2020] | | 61.0 | 81.3 | **86.0** | **88.7** | 48.9 | 67.6 | **72.5** | 75.5 |
| RETROXPERT [Yan et al., 2020] | | 62.1 | 75.8 | 78.5 | 80.9 | 50.4 | 61.1 | 62.3 | 63.4 |
| GRAPHRETRO (ours) | | **63.9** | **81.5** | 85.2 | 88.1 | **53.7** | **68.3** | 72.2 | **75.5** |

by comparing the canonical SMILES of predicted reactants to the ground truth. Atom-mapping is excluded from this comparison, but stereochemistry, which describes the relative orientation of atoms in the molecule, is retained. The evaluation is carried out for two settings, with the reaction class being known or unknown.

**Baselines**  For evaluating overall performance, we compare GRAPHRETRO to nine baselines — four template-based, three template-free, and two semi-template-based methods. These include:

*Template-Based*: RETROSIM Coley et al. [2017b] ranks templates for a given target molecule by computing molecular similarities to precedent reactions. NEURALSYM [Segler and Waller, 2017] trains a model to rank templates given a target molecule. GLN [Dai et al., 2019] models the joint distribution of templates and reactants in a hierarchical fashion using logic variables. DUALTB [Sun et al., 2021] uses an energy-based model formulation for retrosynthesis, with additional parameterizations and loss terms to enforce the duality between forward (reaction prediction) and backward (retrosynthesis prediction). Inference is carried out using reactant candidates obtained by applying an extracted template set to the products.

*Template-Free*: SCROP [Zheng et al., 2019], LV-TRANSFORMER [Chen et al., 2019] and DUALTF [Sun et al., 2021] use the Transformer architecture [Vaswani et al., 2017] to output reactant SMILES given a product SMILES. To improve the validity of their suggestions, SCROP include a second Transformer that functions as a syntax correcter. LV-TRANSFORMER uses a latent variable mixture model to improve diversity of suggestions. DUALTF utilizes additional parameterizations and loss terms to enforce the duality between forward (reaction prediction) and backward (retrosynthesis prediction).

*Semi-Template-Based*: G2GS [Shi et al., 2020] and RETROXPERT [Yan et al., 2020] first identify synthons, and then expand the synthons into reactants by either sequential generation of atoms and bonds (G2GS), or using the Transformer architecture (RETROXPERT). The training dataset for the Transformer in [Yan et al., 2020] is augmented with incorrectly predicted synthons with the goal of learning a correction mechanism.

Results for NEURALSYM are taken from [Dai et al., 2019]. The authors in [Yan et al., 2020] report their performance being affected by the dataset leakage[2]. Thus, we use the most recent results from their website on the canonicalized dataset. For remaining baselines, we directly use the values reported in their paper. For the synthon completion module, we use the *ind* configuration given its better empirical performance.

---

[2]https://github.com/uta-smile/RetroXpert

## 4.1 Overall Performance

**Reaction class unknown** As shown in Table 1, when the reaction class is unknown, GRAPHRETRO outperforms G2Gs by $4.8\%$ and and RETROXPERT by $3.3\%$ in top-1 accuracy. Performance improvements are also seen for larger $n$, except for $n = 5$. Barring DUALTB, the top-1 accuracy is also better than other template-free and template-based methods. For larger $n$, one reason for lower top-$n$ accuracies than most template-based methods is that templates already contain combinations of leaving group patterns. In contrast, our model learns to discover these during training. A second hypothesis to this end is that simply adding log-likelihood scores from edit prediction and synthon completion models may be suboptimal and bias the beam search in the direction of the more dominating term. We leave it to future work to investigate scoring functions that rank the attachment.

**Reaction class known** When the reaction class is known, GRAPHRETRO outperforms G2Gs and RETROXPERT by a margin of $3\%$ and $2\%$ respectively in top-1 accuracy. GRAPHRETRO also outperforms all the template-free methods in top-$n$ accuracy. for GRAPHRETRO are also better than most template-based and template-free methods. When the reaction class is known, RETROSIM and GLN restrict template sets corresponding to the reaction class, thus improving performance. The increased edit prediction performance (Section 4.2) for GRAPHRETRO helps outweigh this factor, achieving comparable or better performance till $n = 5$.

## 4.2 Individual Module Performance

To gain more insight into the working of GRAPHRETRO, we evaluate the top-$n$ accuracy ($n = 1, 2, 3, 5$) of edit prediction and synthon completion modules, along with corresponding ablation studies, with results shown in Table 2.

**Edit Prediction** For the edit prediction module, we compare the true edit(s) to top-$n$ edits predicted by the model. We also consider two ablation studies, one where we directly use the initial edit scores without updating them, and the other where we predict edits using atom-pairs instead of existing bonds and atoms. Both design choices lead to improvements in performance, as shown in Table 2. We hypothesize that the larger improvement compared to edit prediction using atom-pairs is due to the easier optimization procedure, with lesser imbalance between labels 1 and 0.

**Synthon Completion** For evaluating the synthon completion module, we first apply the true edits to obtain synthons, and compare the true leaving groups to top-$n$ leaving groups predicted by the model. We test the performance of both the *ind* and *shared* configurations. Both configurations perform similarly, and are able to identify ~ 97% (close to its upper bound of 99.7%) of the true leaving groups in its top-5 choices, when the reaction class is known. The top-1, 3 and 5 accuracies of the synthon completion for unknown reaction classes for G2Gs are 61.1%, 81.5% and 86.7% respectively, while ours are 75.6%, 92.5% and 96.1%, indicating a 10-14% performance improvement using a classification formulation over the generative one adopted by G2Gs.

Table 2: **Performance Study** of edit prediction and synthon completion modules

| Setting | | Top-$n$ Accuracy (%) | | | | | | | |
|---|---|---|---|---|---|---|---|---|---|
| | | Reaction class known | | | | Reaction class unknown | | | |
| | $n =$ | 1 | 2 | 3 | 5 | 1 | 2 | 3 | 5 |
| Edit Prediction | | **84.6** | **92.2** | **93.7** | **94.5** | **70.8** | **85.1** | **89.5** | **92.7** |
| - without edit score updates | | 84.3 | 92.1 | 93.7 | 94.5 | 70.1 | 84.8 | 89.4 | 92.6 |
| - predicting on atom pairs | | 81.9 | 89.5 | 90.9 | 92.1 | 68.6 | 83.2 | 88.3 | 91.8 |
| Synthon Completion (*ind*) | | **77.4** | 89.5 | **94.2** | **97.6** | **75.6** | 87.4 | 92.5 | 96.1 |
| Synthon Completion (*shared*) | | 76.9 | **89.6** | 93.9 | 97.4 | 74.9 | **87.7** | **92.9** | **96.3** |

## 4.3 Example Predictions

In Figure 2, we visualize the model predictions and the ground truth for three cases. Figure 2a shows an example where the model identifies both the edits and leaving groups correctly. In Figure 2b, the correct edit is identified but the predicted leaving groups are incorrect. We hypothesize this is due to the fact that in the training set, leaving groups attaching to the carbonyl carbon (C=O) are small (e.g. -OH, -NH$_2$, halides). The true leaving group in this example, however, is large. The model is unable to reason about this and predicts the small leaving group -I. In Figure 2c, the model identifies the edit and consequently the leaving group incorrectly. This highlights a limitation of our model. If the edit is predicted incorrectly, the model cannot suggest the true precursors.

## 4.4 Limitations

The simplified and interpretable construction of GRAPHRETRO comes with certain limitations. First, the overall performance of the model is limited by the performance of the edit prediction step. If the predicted edit is incorrect, the true reactants cannot be salvaged. This limitation is partly remedied by our model design, that allows for user intervention to correct the edit. Second, our method is reliant on atom-mapping for extracting edits and leaving groups. Extracting edits directly based on substructure matching currently suffer from false positives, and heuristics to correct for these result in correct edits in only ~90% of the cases. Third, our formulation assumes that we have as many synthons as reactants, which is violated in some reactions. We leave it to future work to extend the model to realize a single reactant from multiple synthons, and introduce more chemically meaningful edit correction mechanisms.

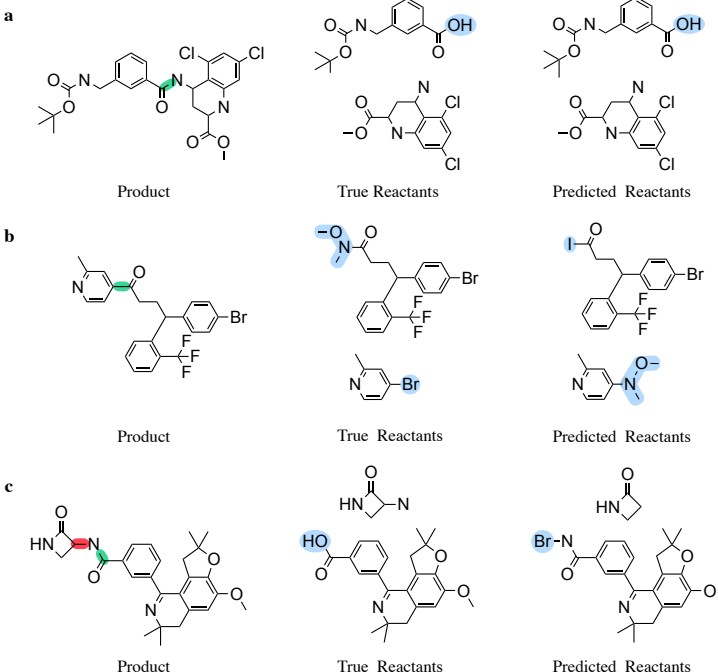

Figure 2: **Example Predictions**. The true edit and incorrect edit (if any) are highlighted in green and red respectively. The true and predicted leaving groups are highlighted in blue. **a**. Correctly predicted example by the model. **b**. Correctly predicted edit but incorrectly predicted leaving groups. **c**. Incorrectly predicted edit and leaving group.

## 5   Conclusion

Previous methods for single-step retrosynthesis either restrict prediction to a template set, are insensitive to molecular graph structure or generate molecules from scratch. We address these shortcomings by introducing a graph-based semi-template-based model inspired by a chemist's

workflow, enhancing the interpretability of retrosynthesis models. Given a target molecule, we first identify synthetic building blocks (*synthons*) which are then realized into valid reactants, thus avoiding molecule generation from scratch. Our model outperforms previous semi-template-methods by significant margins on the benchmark dataset. Future work aims to extend the model to realize a single reactant from multiple synthons, and introduce more chemically meaningful components to improve the synergy between such tools for retrosynthesis prediction and a practitioner's expertise.

## Acknowledgements

This research was supported by the Machine Learning for Pharmaceutical Discovery and Synthesis Consortium at MIT. V.R.S. was also supported by the Zeno Karl Schindler Foundation. C.B. was supported by the Swiss National Science Foundation under the National Center of Competence in Research (NCCR) Catalysis under grant agreement 51NF40 180544. We thank the Leonhard scientific computing cluster at ETH Zürich for providing computational resources.

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
