# Learning Graph Models for Retrosynthesis Prediction

**Vignesh Ram Somnath**[1]      **Charlotte Bunne**[1]      **Connor W. Coley**[2]

**Andreas Krause**[1]    **Regina Barzilay**[3]

[1]Department of Computer Science, ETH
[2]Department of Chemical Engineering, MIT
[3]Computer Science and Artificial Intelligence Lab, MIT
[1]{vsomnath, bunnec, krausea}@ethz.ch, [2]ccoley@mit.edu, [3]regina@csail.mit.edu

## Abstract

Retrosynthesis prediction is a fundamental problem in organic synthesis, where the task is to identify precursor molecules that can be used to synthesize a target molecule. A key consideration in building neural models for this task is aligning model design with strategies adopted by chemists. Building on this viewpoint, this paper introduces a graph-based approach that capitalizes on the idea that the graph topology of precursor molecules is largely unaltered during a chemical reaction. The model first predicts the set of graph edits transforming the target into incomplete molecules called *synthons*. Next, the model learns to expand synthons into complete molecules by attaching relevant *leaving groups*. This decomposition simplifies the architecture, making its predictions more interpretable, and also amenable to manual correction. Our model achieves a top-1 accuracy of 53.7%, outperforming previous template-free and semi-template-based methods.

## 1 Introduction

Retrosynthesis prediction, first formalized by E. J. Corey [7] is a fundamental problem in organic synthesis that attempts to identify a series of chemical transformations for synthesizing a target molecule. In the single-step formulation, the task is to identify a set of reactant molecules given a target. Beyond simple reactions, many practical tasks involving complex organic molecules are difficult even for expert chemists. As a result, substantial experimental exploration is needed to cover for deficiencies of analytical approaches. This has motivated interest in computer-assisted retrosynthesis [6], with a recent surge in machine learning methods [2, 4, 8, 30, 9].

Computationally, the main challenge is how to explore the combinatorial space of reactions that can yield the target molecule. Largely, previous methods for retrosynthesis prediction can be divided into template-based [4, 8, 21] and template-free [2, 30] approaches. Template-based methods match a target molecule against a large set of templates, which are molecular subgraph patterns that highlight changes during a chemical reaction. Despite their interpretability, these methods fail to generalize to new reactions. Template-free methods bypass templates by learning a direct mapping from the SMILES [26] representations of the product to reactants. Despite their greater generalization potential, these methods generate reactant SMILES character by character, increasing generation complexity.

Another important consideration in building retrosynthesis models is aligning model design with strategies adopted by expert chemists. These strategies are influenced by fundamental properties of chemical reactions, independent of complexity level: (i.) the product atoms are always a subset of the

35th Conference on Neural Information Processing Systems (NeurIPS 2021), Sydney, Australia.

**a Edit Prediction**

Product → Synthons

**b Synthon Completion**

Synthons → Reactants

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

| **Template-Based** | | | | | | | | | |
| RETROSIM [4] | | 52.9 | 73.8 | 81.2 | 88.1 | 37.3 | 54.7 | 63.3 | 74.1 |
| NEURALSYM [21] | | 55.3 | 76.0 | 81.4 | 85.1 | 44.4 | 65.3 | 72.4 | 78.9 |
| GLN [8] | | 64.2 | 79.1 | 85.2 | 90.0 | 52.5 | 69.0 | 75.6 | 83.7 |
| DUALTB [23] | | **67.7** | **84.8** | **88.9** | **92.0** | **55.2** | **74.6** | **80.5** | **86.9** |
| **Template-Free** | | | | | | | | | |
| SCROP [30] | | 59.0 | 74.8 | 78.1 | 81.1 | 43.7 | 60.0 | 65.2 | 68.7 |
| LV-TRANSFORMER [2] | | - | - | - | - | 40.5 | 65.1 | 72.8 | 79.4 |
| DUALTF [23] | | **65.7** | **81.9** | **84.7** | **85.9** | **53.6** | **70.7** | **74.6** | **77.0** |
| **Semi-Template-Based** | | | | | | | | | |
| G2GS [22] | | 61.0 | 81.3 | **86.0** | **88.7** | 48.9 | 67.6 | **72.5** | 75.5 |
| RETROXPERT [28] | | 62.1 | 75.8 | 78.5 | 80.9 | 50.4 | 61.1 | 62.3 | 63.4 |
| GRAPHRETRO (ours) | | **63.9** | **81.5** | 85.2 | 88.1 | **53.7** | **68.3** | 72.2 | **75.5** |

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

# A    Message Passing Network

We use the message passing network as proposed in [13], which is an improved variant of the Directed-Message Passing Neural Network (D-MPNN) proposed in [29]. We use this network owing to its good performance across different molecular machine learning tasks.

At message passing step $t$, each bond $(u, v) \in \mathcal{E}$ is associated with two messages $\mathbf{m}_{uv}^{(t)}$ and $\mathbf{m}_{vu}^{(t)}$. Message $\mathbf{m}_{uv}^{(t)}$ is updated using

$$\mathbf{m}_{uv}^{(t+1)} = \mathrm{GRU}\left(\mathbf{x}_u, \mathbf{x}_{uv}, \{\mathbf{m}_{wu}^{(t)}\}_{w \in N(u) \backslash v}\right), \tag{12}$$

where GRU denotes the Gated Recurrent Unit, adapted for message passing [13]

$$\mathbf{s}_{uv} = \sum_{k \in N(u) \backslash v} \mathbf{m}_{ku}^{(t)} \tag{13}$$

$$\mathbf{z}_{uv} = \sigma\left(\mathbf{W_z}\left[\mathbf{x}_u, \mathbf{x}_{uv}, \mathbf{s}_{uv}\right] + b_z\right) \tag{14}$$

$$\mathbf{r}_{ku} = \sigma\left(\mathbf{W_r}\left[\mathbf{x}_u, \mathbf{x}_{uv}, \mathbf{m}_{ku}^{(t)}\right] + b_r\right) \tag{15}$$

$$\tilde{\mathbf{r}}_{uv} = \sum_{k \in N(u) \backslash v} \mathbf{r}_{ku} \odot \mathbf{m}_{ku}^{(t)} \tag{16}$$

$$\tilde{\mathbf{m}}_{uv} = \tanh\left(\mathbf{W}\left[\mathbf{x}_u, \mathbf{x}_{uv}\right] + \mathbf{U}\tilde{\mathbf{r}}_{uv} + b\right) \tag{17}$$

$$\mathbf{m}_{uv}^{(t+1)} = (1 - \mathbf{z}_{uv}) \odot \mathbf{s}_{uv} + \mathbf{z}_{uv} \odot \tilde{\mathbf{m}}_{uv}. \tag{18}$$

After $T$ steps of iteration, we aggregate the messages with a neural network $g(\cdot)$ to derive the representation for each atom

$$\mathbf{c}_u = g\left(\mathbf{x}_u, \sum_{k \in N(u)} \mathbf{m}_{vu}^{(T)}\right). \tag{19}$$

We do not include any stereochemical features, nor is the message passing network stereochemically aware. We leave it to future work to explore the effects of recent message passing networks [19] that utilize aggregation functions that are stereochemically aware.

# B    Leaving Group Attachment

Attaching atoms in the leaving groups were marked during vocabulary construction. The number of such atoms are used to divide leaving groups into single and multiple attachment categories. The single attachment leaving groups are further divided into single and double bond attachments depending on the *valency* of the attaching atom. By default, for leaving groups in the multiple attachments category, a single bond is added between attaching atom(s) on the synthon and leaving groups. For multiple attachment leaving groups with a combination of single and double bonds, the attachment is hardcoded. A single edit can result in a maximum of two attaching atoms for the synthon(s). For the case where the model predicts a leaving group with a single attachment, and the predicted edit results in a synthon with two attaching atoms, we attach to the first atom. For the opposite case where we have multiple attaching atoms on the leaving group and a single attaching atom for the synthon, atoms on the leaving group are attached through respective bonds. The former case represents incorrect model predictions, and is not observed as ground truth.

# C    Experimental Details

Our model is implemented in `PyTorch` [18]. We also use the open-source software `RDKit` [15] to canonicalize product molecules, extracing edits and leaving groups from molecules, for attaching leaving groups to synthons and generating reactant SMILES.

### C.1 Input Features

#### C.1.1 Product & Synthon Graphs

In this graph, the nodes are atoms and bonds are edges. We use the following node and edge features,

| Node Feature | Count | One-hot | Possible Values |
|---|---|---|---|
| Atom symbol | 65 | Yes | C, N, O etc. |
| Atom degree | 10 | Yes | 0, 1, 2, 3, 4, 5, 6, 7, 8, 9 |
| Formal charge of the atom | 5 | Yes | -1, -2, 1, 2, 0 |
| Valency of the atom | 7 | Yes | 0, 1, 2, 3, 4, 5, 6 |
| Hybridization of the atom | 5 | Yes | SP, SP2, SP3, SP3D, SP3D2 |
| Number of associated hydrogens | 5 | Yes | 0, 1, 3, 4, 5 |
| Part of an aromatic ring | 1 | No | 0, 1 |

| Edge Feature | Count | One-hot | Possible Values |
|---|---|---|---|
| Bond type | 4 | Yes | Single, Double, Triple, Aromatic |
| Whether bond is conjugated | 1 | No | 0, 1 |
| Whether bond is part of ring | 1 | No | 0, 1 |

#### C.1.2 Bond Graph

In this graph, the bonds are nodes, and two bonds share an edge if they have a common atom. The features used for this graph include,

| Node Feature | Count | One-hot | Possible Values |
|---|---|---|---|
| Bond type | 4 | Yes | Single, Double, Triple, Aromatic |
| Whether bond is conjugated | 1 | No | 0, 1 |
| Whether bond is part of ring | 1 | No | 0, 1 |

The edge features include the atom features (Node Features table in Appendix C.1.1 for details) of the common atom, the bond type and conjugation of the participating bonds, and if the two bonds are part of the same ring.

### C.2 Dataset Splits

We evaluate our model on the USPTO-50k [20] dataset. We use the same dataset and splits as provided by [8]. The USPTO-50k dataset contains a shortcut in that the product atom with atom-mapping 1 is part of the edit in ~75% of the cases. If the product SMILES is not canonicalized, predictions utilizing operations that depend on the position of the atom or bond will be able to use the shortcut, and overestimate performance. Before extracting edits and leaving groups, we remap the existing dataset to remove this shortcut,

**Remapping USPTO-50k Dataset**    We first canonicalize the product molecule by clearing out atom numbers and converting the molecule to SMILES. The atoms in the canonicalized product have an atom mapping corresponding to their ordering. We then apply substructure matching between the original product and canonicalized product to identify the correspondence between the original and updated atom mappings, which we then use to update the atom mapping of the corresponding reactant atoms. To verify the correctness of the remapping procedure, we compare the number of extracted edits from the original and remapped products and reactants.

### C.3 Hyperparameter Tuning

**Edit Prediction**    The table below indicates the hyperparameter sweep for edit prediction,

| Parameter | Values |
|---|---|
| Hidden dimension of MPN | [256, 512, 768] |
| Hidden dimensions of MLP | [256, 512, [512, 256] |
| MPN depth | [5, 10] |
| Learning rate decay | [0.6, 0.9] |

**Synthon Completion**   We ran a hyperparameter sweep only for the *ind* configuration given its better empirical performance. The sweep configuration is indicated in the following table,

| Parameter | Values |
|---|---|
| Hidden dimensions of MLP | [300, 150, [300, 150]] |

## C.4   Network Architectures

All models are trained with the Adam optimizer and an initial learning rate of $0.001$.

### C.4.1   Edit Prediction

We run the MPN for $T = 10$ iterations, with a hidden layer dimension of 256. The initial edit scores are predicted with a MLP of hidden layer dimension 512. In the reaction class unknown setting, we update the initial edit scores using a smaller MPN which is run for $T = 3$ iterations, and has a hidden layer dimension of 64. We also use dropout on the node embeddings and the hidden layers of MLP with a probability of 0.15 and 0.3 respectively. We apply a learning rate decay of 0.9 based on validation accuracy, with patience 10 and an improvement threshold of 0.01. Gradients are clipped to a norm of 10.0. Both models are trained for 200 epochs. The model has 1.03M parameters in the reaction class known setting, and 1.06M parameters in the reaction class unknown setting.

### C.4.2   Synthon Completion

The network architecture and training details are largely similar across the reaction class known and unknown settings. We run the MPN for $T = 10$ iterations, with a hidden layer dimension of 300. The embedding dimension of leaving groups is set to 200. In the reaction class known setting, the classifier over leaving groups is a two layer MLP, with hidden dimensions of 300 and 150, while in the reaction class unknown setting, the classifier is a single layer MLP with a hidden layer dimension of 300. We also use dropout on the node embeddings and the hidden layers of MLP with a probability of 0.15 and 0.3 respectively. We apply a learning rate decay of 0.9 based on the validation accuracy, with a patience of 5 epochs, and a threshold value for improvement set to 0.01. Gradients are clipped to a norm of 10.0. The model is trained for 100 epochs. When the reaction class is known, the model has 0.84M parameters, while in the reaction class unknown case, the model has 0.81M parameters.

## C.5   Computing

The edit prediction models were trained in about 23-24 hours on a single NVIDIA 1080Ti GPU, while the synthon completion models were trained in about 12-13 hours on the same GPU configuration.

# D   Multiple Edit Prediction

We propose an autoregressive model for multiple edit prediction that allows us to represent arbitrary length edit sets. The model makes no assumption on the connectivity of the reaction centers or the electron flow topology, addressing the drawbacks mentioned in [1, 12].

Each edit step $t$ uses the intermediate graph $\mathcal{G}_s^{(t)}$ as input, obtained by applying the edits until $t$ to $\mathcal{G}_p$. Atom and bond labels are now indexed by the edit step, and a new termination symbol $y_d^{(t)}$ is introduced such that $\sum_{(u,v),k} y_{uvk}^{(t)} + \sum_u y_u^{(t)} + y_d^{(t)} = 1$. The number of atoms remain unchanged during edit prediction, allowing us to associate a hidden state $\mathbf{h}_u^{(t)}$ with every atom $u$. Given representations $\mathbf{c}_u^{(t)}$ returned by the $\mathrm{MPN}(\cdot)$ for $\mathcal{G}_s^{(t)}$, we update the atom hidden states as

$$\mathbf{h}_u^{(t)} = \tau \left( \mathbf{W_h} \mathbf{h}_u^{(t-1)} + \mathbf{W_c} \mathbf{c}_u^{(t)} + b \right). \tag{20}$$

The bond hidden state $\mathbf{h}_{uv}^{(t)} = (\mathbf{h}_u^{(t)} \| \mathbf{h}_v^{(t)})$ is defined similar to the single edit case. We also compute the termination score using a molecule hidden state $\mathbf{h}_m^{(t)} = \sum_{u \in \mathcal{G}_s^{(t)}} \mathbf{h}_u^{(t)}$. The edit logits are predicted by passing these hidden states through corresponding neural networks

$$s_{uvk}^{(t)} = \mathbf{u_k}^T \tau \left( \mathbf{W_k} \mathbf{h}_{uv}^{(t)} + b_k \right) \tag{21}$$

$$s_u^{(t)} = \mathbf{u_a}^T \tau \left( \mathbf{W_a} \mathbf{h}_u^{(t)} + b_a \right) \tag{22}$$

$$s_d^{(t)} = \mathbf{u_d}^T \tau \left( \mathbf{W_d} \mathbf{h}_m^{(t)} + b_d \right). \tag{23}$$

**Training**  Training minimizes the cross-entropy loss over possible edits, aggregated over edit steps

$$\mathcal{L}_e(\mathcal{T}_e) = - \sum_{(\mathcal{G}_p, E) \in \mathcal{T}_e} \sum_{t=1}^{|E|} \left( \sum_{(j,k) \in E[t]} y_{uvk}^{(t)} \log(s_{uvk}^{(t)}) + \sum_{u \in E[t]} y_u^{(t)} \log(s_u^{(t)}) + y_d^{(t)} \log(s_d^{(t)}) \right). \tag{24}$$

Training utilizes *teacher-forcing* so that the model makes predictions given correct histories.

# E   Reaction Class Performance

Table 3 describes the 10 reaction classes found in the USPTO-50k dataset. This description was taken from [4]. Table **??** shows the performance of GRAPHRETRO across the different reaction classes. Reaction class 4 characterizing heterocyle formations is composed only of multiple edit examples, which our model cannot currently predict.

Table 3: **Reaction Class Description** of the 10 reaction classes found in USPTO-50k dataset

| Class | Description | Fraction of Dataset (%) |
|-------|-------------|-------------------------|
| 1 | heteroatom alkylation and arylation | 30.3 |
| 2 | acylation and related processes | 23.8 |
| 3 | CC bond formation | 11.3 |
| 4 | heterocycle formation | 1.8 |
| 5 | protections | 1.3 |
| 6 | deprotections | 16.5 |
| 7 | reductions | 9.2 |
| 8 | oxidations | 1.6 |
| 9 | functional group interconversion (FGI) | 3.7 |
| 10 | functional group addition (FGA) | 0.5 |

Table 4: **GRAPHRETRO Performance** across the USPTO-50k reaction classes

| Model | 1 | 2 | 3 | 4 | 5 | 6 | 7 | 8 | 9 | 10 |
|-------|---|---|---|---|---|---|---|---|---|----|
| GRAPHRETRO | 0.57 | 0.63 | 0.38 | 0.0 | 0.53 | 0.54 | 0.56 | 0.55 | 0.32 | 0.83 |