# OpenReview forum: "Learning Graph Models for Retrosynthesis Prediction"
_NeurIPS.cc/2021/Conference — NeurIPS 2021 Poster_

### Official Review · Reviewer_aDNh · 2021-06-28

**Rating:** 4
**Confidence:** 5

**Summary:**

This paper proposes a new strategy for reactants generation for the retrosynthesis prediction. Motivated by the fact that the graph topology of precursor molecules is largely unchanged during the chemical reaction, the authors propose to expand intermediate synthons into reactants by selecting and attaching leaving groups (molecule sub-graphs extracted from training data) to the synthons. The product molecule is transformed into synthons through graph edits. The proposed method achieves 53.7% top-1 accuracy.

**Limitations And Societal Impact:**

Yes.

**Main Review:**

The major contribution of this paper is a new formulation to complete synthons by selecting leaving groups to obtain the final reactants. While existing methods G2Gs generate reactants by adding atoms one-by-one to the synthons and RetroXpert predicts reactants with a Transformer model from synthons. The advantage of the proposed method is that the reactant generation process is simplified and therefore can achieve better accuracy than G2Gs and RetroXpert. However, the related method RetroPrime [1] is not compared in this paper, which also belongs to the semi-template methods and it achieves 64.8% top-1 accuracy while the proposed method 63.9%. This makes the proposed method less significant. I am worried that the contribution of this paper is incremental and not significant enough.

Line #48 Is the atom-mapping bias first found by the authors, any reference to the shortcut removing method?

Line #130 "95% of the training set consists of single edit examples the remainder of this section describes the setup for single edit
 prediction", line #150 proposes to leverage the dependencies of multiple edits, it seems contradictory.

Line #152 "build a graph with bonds (u, v) as nodes, and introduce an edge between bonds sharing an atom" sounds interesting, is this originally proposed by the authors?

Could the authors also provide experimental results on the larger dataset USPTO-full as GLN to demonstrate the scalability of the proposed method? I am worried that the proposed leaving groups attachment method can only be applied to small datasets and is not generalizable.


[1] Wang, Xiaorui, et al. "RetroPrime: A Diverse, plausible and Transformer-based method for Single-Step retrosynthesis predictions." Chemical Engineering Journal 420 (2021): 129845.



**Time Spent Reviewing:**

5

---

> ### Author Response · Authors · 2021-08-10
> **General Comments and Large-Scale Dataset Experiments**
>
> Thank you very much for your insightful comments!
>
> > The related method RetroPrime [1] is not compared in this paper, which also belongs to the semi-template methods and it achieves 64.8% top-1 accuracy.
>
> Thank you for the reference, we have updated our table accordingly. Indeed, *RetroPrime* outperforms our model on the top-1 accuracy in the reaction class known setting. However, our model achieves better top-1 accuracy performance (+2.3\%) on the harder, more practical scenario of unknown reaction classes. We hypothesize that the marginal improvements in reaction class known setting could arise from the re-ranking scheme employed by *RetroPrime* after beam search to deal with invalid SMILES. To maintain a consistent evaluation with previous semi-template methods, we directly use the outputs of beam search to calculate the overall performance but acknowledge that this re-ranking can be useful even for our model to help address some of the issues of beam search as mentioned in Section 4.1
>
> > Line #48 Is the atom-mapping bias first found by the authors, any reference to the shortcut removing method?
>
> We did notice the atom-mapping bias first but had to wait for the updated results from the authors of [RetroXpert](https://github.com/uta-smile/RetroXpert) to include in our paper.
>
> > Line #130 "95% of the training set consists of single edit examples the remainder of this section describes the setup for single edit prediction", line #150 proposes to leverage the dependencies of multiple edits, it seems contradictory.
>
> Thank you for your comment. We only wanted to highlight that using a single-edit model was informed by the characteristics of the USPTO-50k dataset, and one can always use a multi-edit or atom-pair-based model for datasets where the characteristic is not satisfied, and in no way is a limitation of the model formulation. We will make sure to clarify this in the revised version.
>
> > Line #152 "build a graph with bonds (u, v) as nodes, and introduce an edge between bonds sharing an atom" sounds interesting, is this originally proposed by the authors?
>
> To the best of our knowledge, we have not seen this idea being used in any papers around molecular machine learning. Reviewer okuQ pointed us to a paper using line graphs for link prediction, which we were not aware of, but have cited in our paper now.
>
> > Could the authors also provide experimental results on the larger dataset USPTO-full as GLN to demonstrate the scalability of the proposed method?
>
> Given the limited time for the rebuttal, we were not able to run experiments for the full model, but instead, report results for synthon completion. There are 10000 unique leaving groups in the full dataset, of which more than half only have a single occurrence. The test set coverage also drops to ~95%. Using default hyperparameters, our model was able to achieve 66.4% synthon completion performance, comparable to the USPTO-50k performance. The performance can be further improved by training for longer and more hyperparameter tuning. Unlike USPTO-50k, only 70% of the examples are single-edits. Hence, we need to use atom pairs for edit prediction, which has been shown to achieve good performance on reaction classification tasks (Jin et al. NeurIPS 2017, Coley et al. 2019). Putting the above results and references together, we expect no scalability issues for our model. We do acknowledge the need to ensure the code for attaching leaving groups to synthons works as expected for the new leaving groups.

---

> > ### Comment · Reviewer_aDNh · 2021-08-12
> > **"build a graph with bonds (u, v) as nodes" is similar to D-MPNN**
> >
> > Thanks for the authors' detailed response!
> >
> >
> > >>Line #152 "build a graph with bonds (u, v) as nodes, and introduce an edge between bonds sharing an atom" sounds interesting, is this originally proposed by the authors?
> >
> > >To the best of our knowledge, we have not seen this idea being used in any papers around molecular machine learning. Reviewer okuQ pointed us to a paper using line graphs for link prediction, which we were not aware of, but have cited in our paper now.
> >
> > I just found out that D-MPNN [1] proposed a bond-level message passing to learn molecular presentation. It seems like the proposed bond-level message passing in D-MPNN is the same to build a graph with bonds as nodes here. Could the authors clarify the major difference? Please the authors consider including proper citation to the related work D-MPNN.
> >
> >
> > [1] Yang, Kevin, et al. "Analyzing learned molecular representations for property prediction." Journal of chemical information and modeling 59.8 (2019): 3370-3388.

---

> > > ### Author Response · Authors · 2021-08-16
> > > **Clarification of the Line-Graph Model**
> > >
> > > > I just found out that D-MPNN [1] proposed a bond-level message passing to learn molecular presentation. It seems like the proposed bond-level message passing in D-MPNN is the same to build a graph with bonds as nodes here. Could the authors clarify the major difference? Please the authors consider including proper citation to the related work D-MPNN.
> > >
> > > In this work, we use the MPNN by Jin et al. (2019), which we notice builds upon work mentioned by the reviewer (Yang et al. (2019)). We will of course add the citation to our paper for completeness.
> > > While the MPNN by Yang et al. (2019) does not build a line graph explicitly, it updates the messages, instead of hidden representation of the nodes, based on incoming messages into the node of interest and thus indeed shares some similarities to our line-graph based work. Yang et al., however, compute hidden representations for each edge of the directed graph rather than for each bond. Ultimately, no representation on the bond level is computed (see [code](https://github.com/chemprop/chemprop/blob/master/chemprop/models/mpn.py#L118)). The construction of the line graph is informed by our edit correction module, to utilize the dependencies between bond edits and correct initial edit scores. Besides directly parameterizing the object of interest, the corresponding line graph has fewer edges, requiring fewer messages to achieve the bond representations.

---

> > > > ### Comment · Reviewer_aDNh · 2021-08-16
> > > > **Clarification of the Line-Graph Model**
> > > >
> > > > Thank the authors for the clarification.

---

> > ### Comment · Reviewer_aDNh · 2021-08-12
> > **Generalization on Large-Scale Dataset**
> >
> > Thanks for the further study on the large dataset experiment!
> >
> > The authors acknowledge "using a single-edit model was informed by the characteristics of the USPTO-50k dataset", and it seems like the proposed method exploits the bias of the USPTO-50k dataset (95% of the training examples are single-edit). Thank the authors for providing initial investigation results on the full dataset. The authors expect no scalability issues by providing some analysis of the full dataset (70% of examples are single-edit), synthon completion result, and some references. However, the generalization is still not demonstrated by experiments.
> >
> > Besides, after reading the source code of the implementation (thanks for providing the code), I found there were a lot of manual effort and hard coding to fix/edit the molecule. And this part was not mentioned in the paper. This makes me further doubt the generalization of the proposed method since the situation will be more complicated when working on the full dataset.
> >
> > I would suggest the authors provide the final experimental results on the full dataset and may re-submit to the following top conferences like AAAI, ICLR, and ICML if more time is needed.

---

> > > ### Author Response · Authors · 2021-08-16
> > > **Leaving Group Attachment and Scalability**
> > >
> > > Thank you for your comments!
> > >
> > > > The authors acknowledge "using a single-edit model was informed by the characteristics of the USPTO-50k dataset", and it seems like the proposed method exploits the bias of the USPTO-50k dataset (95% of the training examples are single-edit). Thank the authors for providing initial investigation results on the full dataset. The authors expect no scalability issues by providing some analysis of the full dataset (70% of examples are single-edit), synthon completion result, and some references. However, the generalization is still not demonstrated by experiments.
> > >
> > > As shown in the ablation studies, we evaluate both the single-edit and atom-pair model, and the single-edit model performs better. We use his model because of its better empirical performance on the validation set, which we hypothesize is largely due to the single-edit characteristic of the USPTO-50k dataset. A similar process of evaluating both models will also be done on the large dataset before choosing the final model. This is no different from hyperparameter tuning or architecture search on the validation set which is standard practice.
> > >
> > > > Besides, after reading the source code of the implementation (thanks for providing the code), I found there was a lot of manual effort and hard coding to fix/edit the molecule. And this part was not mentioned in the paper. This makes me further doubt the generalization of the proposed method since the situation will be more complicated when working on the full dataset.
> > >
> > > We analyzed the statistics of the leaving groups on the larger dataset and it indeed confirms our assumptions made regarding redundancies of leaving groups. Thus, we argue that our method is scalable. On the larger dataset, 500 leaving groups are sufficient to cover 93% of the examples on the training set (by extension, validation, and test set since the split is random). 60% of these groups are single attachment groups (as described below), and the remaining 40% have few differing atoms from leaving groups for which we already covered the multi-attachment procedure. Considering the top performance on the dataset is in the mid-40s, having coverage of 93% is a reasonable assumption to work with.
> > >
> > > The leaving group attachment process is described in more detail in Appendix B. The goal is to best ensure the chemical correctness of the final molecule and capture as much complexity as possible with limited manual effort (in accordance with current literature on graph-based reaction classification). The leaving groups are first categorized into single or multiple attachment types depending on the number of participating atoms (marked during vocabulary construction). This categorization can be accomplished by simply looping over the leaving groups and counting the number of attaching atoms.
> > >
> > > These groups can further be subdivided into single bond (default) and double bond-based attachments, and cyclic attachments, by applying valency constraints following the octet rule, and inferring the type of possible bonds based on the valency and charges (if any). An oxygen atom, for example, has a possibility of a double bond, but an oxygen atom with a negative charge forms a single bond.
> > >
> > > The procedure described above incorporates basic chemical rules for attaching leaving groups, something that is currently missing from the RDKit software suite. In the multi-attachment category, the participating atoms for some leaving groups attach to a single synthon atom, while for some other leaving groups, the attachment happens to multiple synthon atoms. Considering edge cases like these is good practice in software engineering and helps improve the completeness of the attachment.
> > >
> > > The final loop over the generated reactant atoms takes care of the number of hydrogens attached to each atom. If a carbon atom for example has 4 bonds already, and the number of hydrogen atoms (explicit or implicit) is not zero, then that is an invalid setting and the number of hydrogen atoms is then set to zero. This has already been done for [reaction classification]((https://github.com/connorcoley/rexgen_direct/blob/master/rexgen_direct/scripts/eval_by_smiles.py) across the larger USPTO-480k dataset, and we use this code in our work, too.

---

> > ### Author Response · Authors · 2021-08-16
> > **RetroPrime vs. GraphRetro**
> >
> > > The related method RetroPrime [1] is not compared in this paper, which also belongs to the semi-template methods and it achieves 64.8% top-1 accuracy while the proposed method 63.9 (reaction class known)%. This makes the proposed method less significant.
> >
> > Including an additional note on the comparison with RetroPrime, these are two different modeling approaches. RetroPrime utilizes two Transformers to accomplish the products-to-synthons and synthons-to-reactant transformations. This allows some corrective ability (i.e., incorrect synthons can still be translated to valid reactants), but the approach requires heavy (10x) data augmentation and it is harder to introduce manual corrections. On the other hand, our formulation is rooted in utilizing conserved topologies in the transformation from products to reactants, which achieves comparable or better top-1 accuracy on the unknown reaction class setting, eliminates the need for data augmentation, is easier to correct manually (example correct the edit suggested or use one of multiple suggestions), and is more naturally aligned with a chemist's intuition. However, the lack of an automated way to correct edits makes the overall performance reliant on the edit prediction, which is also addressed in the limitations section.

---

### Official Review · Reviewer_okuQ · 2021-07-07

**Rating:** 5
**Confidence:** 5

**Summary:**

This paper deals with molecular graph generation via synthon, incomplete molecular graphs.
Unlike existing works, this paper formulates the completion of incomplete molecular graphs not as a generation problem but as a classification problem to select one of a set of pre-computed leaving groups fragments.
The training and evaluation are performed using the standard dataset USPTO-50k. The proposed model has high retrosynthesis performance, covering a large number of responses with a small set of leaving groups.

**Ethical Concerns:**

No specific concerns.

**Limitations And Societal Impact:**

No specific opinion.

**Main Review:**

## Originality:

Graph retrosynthesis based on synthon was proposed by [20], and this paper is its follower. Therefore, this paper does not propose a new paradigm to the retrosynthesis problem.
However, unlike [20, 26], which has been sticking to the graph generation model, the switch to the problem of identifying the missing parts is novel.

Recently AIzynthfinder has become a hot topic as a free and powerful retrosynthesis engine.
Even if NeurIPS is purely a place for research in machine learning theory (which I don't think it is), I think retrosynthesis works should cite and comment on this engine.

- Thakkar A, Kogej T, Reymond J-L, et al (2019) Datasets and their influence on the development of computer assisted synthesis planning tools in the pharmaceutical domain. Chem Sci. https://doi.org/10.1039/C9SC04944D
- Genheden S, Thakkar A, Chadimova V, et al (2020) AiZynthFinder: a fast, robust and flexible open-source software for retrosynthetic planning. J. Cheminf. https://jcheminf.biomedcentral.com/articles/10.1186/s13321-020-00472-1

## Quality:
Technical contents are sound. In Sec. 4.4, the limitations are analyzed in a detailed manner. this is a plus.

## Clarity:
This paper lacks an explanation of the background knowledge of the retrosynthesis problem. Therefore, it may be almost incomprehensible for readers who have no prior knowledge of this problem.
In addition, some readers may find it difficult to understand the network architecture (Eqs. (2-8)), because the intuition why this configuration was chosen is not described.

## Significance:
Classification problems are generally less complex than generative problems. Therefore, I expected good experimental results.
Unfortunately, the reported results are not clearly superior to the seminal work of [20].

The graph-to-graph method needs to perform atom mapping in advance. On the other hand, the SMILES-focused template-free method does not require such analysis. Table 1 also shows that DualTF [21], which is a template-free method, outperforms synthon-based methods including the proposed method, especially in class unknown cases.

Therefore, I have to say that it is difficult to convincingly argue the necessity of the novel proposal to solve the retrosynthesis by leaving group classification.

(+) novel classification formulation of retrosynthesis
(-) lack of explanations in basic retrosynthesis problem
(--) experimental results are weak

## After author feedbacks

First of all, I would like to thank the authors for their efforts for detailed feedbacks.
Feedbacks resolve some of my concerns.

At the same time, I still have a concern about experimental performances. As explained in the submitted manuscript and the feedbacks, the performance of the entire retrosynthesis is largely affected by the edit prediction. I understand the effectiveness of the proposed framework for the synthon completion subproblem, but its impact on the main retrosynthesis problem is mixed: sometimes win and sometimes lose against the G2Gs baseline.

Given these, my final evaluation score is weak reject (+1). Thank you.


**Time Spent Reviewing:**

4

---

> ### Author Response · Authors · 2021-08-10
> **Clarification on Experimental Results**
>
> Thank you very much for your insightful comments. We have added the suggested citations.
>
> > Classification problems are generally less complex than generative problems. Therefore, I expected good experimental results. Unfortunately, the reported results are not clearly superior to the seminal work of [20].
>
> Classification problems indeed are generally less complex than generative problems. The top-1, 3, 5 accuracies of the synthon completion for unknown reaction classes in [20] are 61.1%, 81.5% and 86.7% respectively, while ours are 75.6%, 92.5% and 96.1%, indicating a **10-14% performance improvement** over the generative model.
> Additionally, G2Gs concatenates the embeddings of atoms in the atom pair before predicting the edit score. In our experiments, we noticed that such concatenation helps **improve edit prediction performance by 3-4%** between the original dataset with the shortcut and the new dataset free of this shortcut. The authors of G2Gs do not make their code publicly available for us to verify if they used the shortcut-free dataset, nor are any comments mentioned in the paper. Despite this, our top-1 accuracy is almost **5% higher than theirs** in the harder unknown reaction class setting. (Comments continued in next point)
>
> > Table 1 also shows that DualTF [21], which is a template-free method, outperforms synthon-based methods including the proposed method, especially in class unknown cases.
>
> As mentioned in the limitations and experiments sections, overall model performance is heavily reliant on edit prediction, and the resulting search space can be biased by the incorrectly predicted edits and their scores, not allowing us to fully exploit the performance of synthon completion. The duality enforcement boosts the model performance as shown by the authors of DualTF, and this is a general idea applicable to our model (also noted by reviewer bGnx). However, using it directly would not allow us to completely operate in the graph paradigm, and we leave it to future work to explore this direction. Synthon-based methods make the formulation more transparent and aligned with a chemist's intuition, and the small loss in performance could be traded off against these other practically desirable attributes.
>
> > This paper lacks an explanation of the background knowledge of the retrosynthesis problem. Therefore, it may be almost incomprehensible for readers who have no prior knowledge of this problem. In addition, some readers may find it difficult to understand the network architecture (Eqs. (2-8)), because the intuition why this configuration was chosen is not described.
>
> Thank you for your comment, we will improve the problem introduction in Sections 1 and 3 in the revised version. Regarding Eq. (2), we use the message-passing network (MPN) as described because of its good performance across different tasks in molecular machine learning. Eq.(3) and Eq. (4) are standard MLPs. For Eq. (5-8), we attempt to leverage the dependencies that exist between the atom and bond edits, and use an MPN on this dependency graph to correct the initial edit scores, similar to how LSTMs work.
>
> We would like to reiterate that the main contribution of this work is a semi-template based method for retrosynthesis prediction that operates at the level of conserved subgraphs, accomplishing the transformation from products to reactants by first identifying synthons and then completing synthons into valid reactants by using subgraphs (leaving groups) from a precomputed vocabulary. This model formulation aligns with a chemist's intuition about the process, improving model transparency and providing a desirable attribute in practice. Furthermore, utilizing leaving groups instead of character-by-character or atom-based generative models allows us to maintain reasonableness of solutions (as shown by the increased performance of synthon completion). Leaving groups and their synthesis properties are also well studied further improving our model's practical applicability.
>
> This formulation allows us to achieve better top-1 accuracy than previous semi-template-based methods, while achieving comparable top-1 accuracies to template-free methods and most template-based methods. We do acknowledge the reliance of overall performance on edit prediction, and improvements in this step by introduction of correction mechanisms for e.g. (as we attempt with the MPN on the dependency graphs) would directly translate into larger improvements on the overall performance.

---

> > ### Comment · Reviewer_okuQ · 2021-08-15
> > **Thank you for your comment**
> >
> > Authors, thank you for your comments.
> > Especially, the content of the third comment (background and equation explanations) would be helpful for many unfamiliar readers.

---

> > > ### Author Response · Authors · 2021-08-16
> > > **Thanks!**
> > >
> > > We are glad that our comments clarified your questions! We will of course add these additional explanations to the paper.

---

### Official Review · Reviewer_Az8s · 2021-07-15

**Rating:** 8
**Confidence:** 5

**Summary:**

The paper proposes an elegant model for stepwise retrosynthesis prediction via retrons and synthons using a graph neural network based model and symbolic graph manipulation.

Strong results are achieved.

**Limitations And Societal Impact:**

Yes, the limitations are clearly and openly described.

**Main Review:**

The authors present a neural-symbolic algorithm that performs retrosynthetic step prediction by using graph neural networks to predict symbolic graph edits. This is done in a very elegant way, using a two step procedure inspired by how chemists learn to perform retrosynthesis: By identifying reaction centers (graph edits) and then filling in the missing components. This is done using a vocabulary of synths instead of letting a graph generative model freely adding in atoms and bonds, which is very important for maintaining reasonableness. This paper is a very important step forward for computer aided synthesis.

This reviewer would recommend to accept the paper at Neurips, but would encourage the authors to add some extra info (see below)

positives:
- the model is very well motivated, elegant, and strongly maps to the domain.
- the use of edit prediction reduces the complexity to O(N) instead of O(N^2) for atom pair prediction.
- The presentation is very clear
- the validation uses an established dataset
- Some limitations remain, but the authors clearly highlight how they can be solved in future work.
- Related work is well acknowledged.
= code is provided and reads well (no time to run it unfortunately)

negatives:
- a breakdown for performance across different reaction classes would be desirable
- the description of stereochemistry handling could be improved


**Time Spent Reviewing:**

3

---

> ### Author Response · Authors · 2021-08-10
> **Reaction Class Performance and Stereochemistry Handling**
>
> Thank you for the feedback and insightful comments!
>
> The following table shows the top-1 accuracy of the model across different reaction classes:
>
> | Model      | 1    | 2    | 3    | 4   | 5    | 6    | 7    | 8    | 9    | 10   |
> |------------|------|------|------|-----|------|------|------|------|------|------|
> | GraphRetro | 0.57 | 0.63 | 0.38 | 0.0 | 0.53 | 0.54 | 0.56 | 0.55 | 0.32 | 0.83 |
>
> Reaction class 4 is composed of only multiple edits, which our current single-edit model does not predict. We will add a similar table for atom-pair based model in the updated version.
>
>
> > Stereochemistry Handling
>
> Our model currently does not incorporate any stereochemical features, and the message passing layers are stereochemically invariant. During evaluation, we maintain the same stereochemistry when going from products to reactants. We leave it to future work to explore the effects of recent message passing networks (Pattanaik et al. 2019) that utilize aggregation functions that are stereochemically aware.

---

### Official Review · Reviewer_bGnx · 2021-07-16

**Rating:** 6
**Confidence:** 3

**Summary:**

This paper proposes a new retrosynthetic prediction model using neural networks. The algorithm proposed to make the prediction based on breaking the target product into synthons and adding leaving groups to the synthons. The main difference of the proposed algorithm to the existing synthon-based retrosynthetic model is on introducing the vocabulary of leaving groups to complete the synthons in an efficient way.  Extensive evaluation verifies that the proposed method is competitive with the existing algorithms, and each algorithmic component plays a meaningful role.

**Limitations And Societal Impact:**

The authors adequately addressed the limitations and potential negative societal impact of their work.

**Main Review:**

This paper proposes a solid idea to using leaving groups to simplify the procedure of completing the synthons obtained from the target molecule. Additional neural architecture to better predicts the bond edit score seems like a useful trick. Although the performance is not strictly state-of-the-art (when compared to DualTF), I think the proposed idea is meaningful enough to be published at the NeurIPS conference.

Comments and questions:
- It seems that the proposed method underperforms when compared to DualTF although it uses an additional knowledge of synthons. Still, I do not think this is a critical weakness since the DualTF idea (enforcing duality) is quite general and applicable to the proposed model.
- I suggest the authors to make reference to [Cai et al., 2020] who used line graphs to make link prediction using graph neural network. The line graph is exactly the same graph used for updating bond edit scores, i.e., a graph with bond as nodes and edges between bonds sharing an atom.
- Title and naming (GraphRetro) of the algorithm can be improved. They do not give much information about the proposed idea.

**Time Spent Reviewing:**

2

---

> ### Author Response · Authors · 2021-08-10
> **Performance Comparison to DualTF and Additional References**
>
> We thank the reviewer for their insightful feedback and comments!
>
> > It seems that the proposed method underperforms when compared to DualTF although it uses an additional knowledge of synthons. Still, I do not think this is a critical weakness since the DualTF idea (enforcing duality) is quite general and applicable to the proposed model.
>
> As mentioned in Sections 4.1 and 4.4, the overall performance is reliant on edit prediction and the resulting search space can be biased by the incorrectly predicted edits and their scores, not allowing us to fully exploit the performance of synthon completion. The duality enforcement boosts the model performance as shown by the authors of DualTF, and we agree this is a general idea applicable to our model. However, using it directly would not allow us to completely operate in the graph paradigm, and we leave it to future work to explore this direction.
>
> > I suggest the authors to make reference to [Cai et al., 2020] who used line graphs to make link prediction using graph neural network. The line graph is exactly the same graph used for updating bond edit scores, i.e., a graph with bonds as nodes and edges between bonds sharing an atom.
>
> Thank you for the reference. We were not aware of this work, and indeed it bears similarities to our update module. We will cite it in the revised version.
>
> > Title and naming (GraphRetro) of the algorithm can be improved. They do not give much information about the proposed idea.
>
> Thank you for the suggestion, we will be sure to incorporate this in the revised version.

---

### Decision · Program_Chairs · 2021-09-28

**Decision:**

Accept (Poster)

**Comment:**

This paper proposes a new method, GraphRetro, for single step retrosynthesis. GraphRetro treats retrosynthesis as a two stage problem. The first stage predicts the reaction edit, which is similar to previous methods. The second one predicts which leaving group should be attached to the synthons. Different from existing methods, this paper treats the second stage as a multi-class classification problem, which is simpler and achieves some better top-k accuracy for k<5. However, this paper should compare GraphRetro with G2Gs more sufficiently and provide more insights. Besides, this paper should provide large scale experiments on USPTO-full.

**Consistency Experiment:**

NeurIPS has a long history of experimentation. In 2014, NeurIPS ran an experiment in which 10% of submissions were reviewed by two independent committees to quantify the randomness in the review process. This year, we repeated a variant of this experiment to see how the quality of the review process has changed over time.  This paper was part of the experiment and was therefore assigned to two committees (consisting of reviewers, an Area Chair, and a Senior Area Chair) that reached independent decisions.  If both committees made the same recommendation, this recommendation was followed. If a single committee recommended acceptance, the paper was accepted (with the exception of a few cases in which the other committee identified what we considered a fatal flaw, e.g., an error in a key result).

This copy’s committee reached the following decision: **Accept (Poster)**

The other committee assigned to the paper recommended **Reject**.  You can find the other set of reviews, along with any follow up discussion with the authors here:
https://openreview.net/forum?id=LyjH88yV7F